

# Novel roles of $\kappa$-opioid receptor in myocardial ischemia-reperfusion injury

Wen Zhang[1,2], Qi Zhang[3], Yali Liu[1], Jianming Pei[1] and Na Feng[1]

[1] Department of Physiology and Pathophysiology, Fouth Military Medical University, Xi'an, Shaanxi, China
[2] School of Life Science, Northwest University, Xi'an, Shaanxi, China
[3] Graduate School, Dalian Medical University, Dalian, Liaoning, China

## ABSTRACT

Acute heart attack is the primary cause of cardiovascular-related death worldwide. A common treatment is reperfusion of ischemic tissue, which can cause irreversible damage to the myocardium. The number of mitochondria in cardiomyocytes is large, which generate adenosine triphosphate (ATP) to sustain proper cardiac contractile function, and mitochondrial dysfunction plays a crucial role in cell death during myocardial ischemia-reperfusion, leading to an increasing number of studies investigating the impact of mitochondria on ischemia-reperfusion injury. The disarray of mitochondrial dynamics, excessive $Ca^{2+}$ accumulation, activation of mitochondrial permeable transition pores, swelling of mitochondria, ultimately the death of cardiomyocyte are the consequences of ischemia-reperfusion injury. $\kappa$-opioid receptors can alleviate mitochondrial dysfunction, regulate mitochondrial dynamics, mitigate myocardial ischemia-reperfusion injury, exert protective effects on myocardium. The mechanism of $\kappa$-OR activation during myocardial ischemia-reperfusion to regulate mitochondrial dynamics and reduce myocardial ischemia-reperfusion injury will be discussed, so as to provide theoretical basis for the protection of ischemic myocardium.

## INTRODUCTION

Cardiovascular disease is a notable global issue in terms of public health. According to an updated summary, cardiovascular diseases rank first in terms of disease-related deaths with the incidence of cardiovascular diseases continues to increase year by year among Chinese residents. Ischemic heart disease stands as one of the primary causes of cardiovascular mortality in China (*The Writing Committee of the Report on Cardiovascular Health and Diseases in China & Hu, 2023*). The most effective treatment for this disease is to restore the blood supply of the coronary arteries, while the ischemic myocardium will still suffer from persistent injury after restoring blood flow and reperfusion, which is called myocardial ischemia-reperfusion injury (MI/RI) (*Algoet et al., 2023*). At present, there are many therapeutic measures for myocardial ischemia-reperfusion injury, such as percutaneous coronary intervention (PCI), targeted temperature, and hernia. Direct intra-coronary cooling has a potentially beneficial effect in a porcine model of myocardial ischemia-reperfusion (*Kim et al., 2005*). In addition, *Roehl et al. (2013)* reported that xenon

Corresponding authors
Jianming Pei, jmpei8@fmmu.edu.cn
Na Feng, cxnna@163.com

limited progressive adverse cardiac remodeling and systolic dysfunction at 28 days after perioperative myocardial infarction.

Ischemia-reperfusion injury is a pathophysiological mechanism that occurs in many organs (such as the brain, heart, and liver). As early as 60 years ago, it was found that reperfusion can cause a series of myocardial pathological changes, which are mainly about/mitochondrial dysfunction, oxidative stress, apoptosis, metabolic disorders, and inflammatory response (*Jennings et al., 1960*). Mitochondria are organelles known for their high level of activity, which constantly undergo fusion and fission *in vivo*, and the ongoing fusion and fission process to achieve dynamic balance, referred to as mitochondrial dynamics. According to recent research, mitochondrial dysfunction from mitochondrial dynamics disorders is closely associated the pathogenesis of such as MI/RI (*Zhou et al., 2021*). Mitochondrial fusion and fission become imbalanced, which increasing apoptosis, and resulting irreversible damage to the myocardium during MI/RI (*Paradies et al., 2018*). In cardiac myocytes, opioid receptors are abundant, and myocardial ischemic preconditioning increases opioid peptide release, then activates opioid receptors, improves mitochondrial dysfunction at the onset of injury, and attenuates MI/RI (*Fraessdorf et al., 2015*). According to our previous studies, the activition of the $\kappa$-opioid receptor ($\kappa$-OR), effectively safeguards the ischemia-reperfusion (I/R) myocardium (*Cheng et al., 2007*). In this study, based on our research, we discussed and elaborated the regulation of $\kappa$-OR on mitochondria when MI/RI occurs, providing a theoretical reference for MI/RI research.

## OPIOID PEPTIDES AND THEIR RECEPTORS IN MYOCARDIAL PROTECTION

Opioid receptor (OR) is widely distributed throughout the body and belongs to the G-protein-coupled receptors (GPCR) family, which consists of seven-transmembrane proteins (*Vizurraga et al., 2020*). ORs are mainly divided into three subtypes: $\mu$($\mu$-OR), $\delta$ ($\delta$-OR), and $\kappa$ ($\kappa$-OR) (*Lesnik et al., 2021*). ORs can be activated by endogenous opioid peptides and exogenous opioids, exert cell signal transduction through the G protein signalling pathway, and regulate various physiological activities, such as body movement, respiration, temperature, mood, and pain response (*Badal et al., 2018*).

Cardiomyocytes can synthesize, store, and release opioid receptor polypeptides (*Barron, Jones & Caffrey, 1995*). Earlier studies have shown that specific activation of the preischemic $\kappa$b receptor can abnormally increase myocardial infarction size and arrhythmia (*Wong, Lee & Tai, 1990*; *Chien et al., 1994*; *Yu et al., 1999*). Recent evidence suggests that opioids are able to prevent necrosis and apoptosis of cardiomyocytes during I/R and improve cardiac contractility in the reperfusion period, and the OR agonists exert an infarct-reducing effect with prophylactic administration (*Maslov et al., 2016*) Current studies have found that $\kappa$-OR and $\delta$-OR are primarily expressed in cardiomyocytes, and whether $\mu$-OR exists remains controversial (*Headrick et al., 2015*). $\kappa$-OR plays an important role in cardiovascular protection. Studies have reported that $\kappa$-OR is involved in ischemic cardioprotection, prevention of arrhythmia, and its selective agonist U50,488H can prevent MI/RI (*Guo*

*et al., 2011*; *Peart et al., 2008*; *Skrabalova et al., 2012*).Our group has been studying the protective effect of activating $\kappa$-OR on the myocardium.It has been found that activating $\kappa$-OR during MI/RI can significantly reduce myocardial infarct size, inhibit apoptosis and reduce the incidence of arrhythmia, activating $\kappa$-OR during heart failure can regulate cardiac function in rats, activating $\kappa$-OR can relax blood vessels and reduce blood pressure in a spontaneously hypertensive rat model, activating $\kappa$-OR can reduce pulmonary vascular remodeling and right ventricular hypertrophy inhypoxic pulmonary hypertension rats. In a word, activating $\kappa$-OR plays a clear protective role in the cardiovascular system (*Tong et al., 2016*; *Zhang et al., 2018*; *Shi et al., 2013*; *Wang et al., 2020*). In addition, it has been found that $\delta$-OR can also play a protective role in MI/RI myocardium, morphine can reduce myocardial mitochondrial injury, regulate mitophagy, and reduce myocardial apoptosis after hypoxia/reoxygenation (H/R) in cardiac H9C2 cells through activating of epidermal growth factor by $\delta$-OR (*Xu et al., 2022*). Additionally, up-regulation of $\delta$-OR can reduce I/R-induced myocardial infarct size which can safeguard against I/R-induced myocardial mitochondrial impairment. Furthermore, morphine has the potential to decrease myocardial damage through the regulation of autophagy (*Xu et al., 2021*).

## MECHANISM OF ACTIVATING $\kappa$-OR IN RELIEVING MYOCARDIAL ISCHEMIA-REPERFUSION INJURY

### Activating $\kappa$-OR regulates mitochondrial function anti-MI/RI

#### *Mitochondria and MI/RI*

Mitochondria are dynamic organelles responsible for energy production. And play an important role in cellular metabolism, regulating key mechanisms of metabolic homeostasis, immune responses, and emergency signalling. Mitochondrial dysfunction compromises tissue integrity and is associated with various human diseases (*Wang et al., 2022a*; *Wang et al., 2022b*). Mitochondrion covered by a double membrane organelle found in most cells which is an energy-producing structure in the cell. Mitochondria are involved in the tricarboxylic acid cycle, and fatty acid oxidation, and regulate cell cycle and apoptosis *in vivo*. In cardiac myocytes, mitochondria are very abundant, and under physiological conditions, mitochondria require large amounts of oxygen for oxidative phosphorylation to produce ATP to provide energy to cells. During ischemia, the oxygen and nutrient supply of myocardial cells is insufficient, cellular metabolism changes from mitochondrial oxidative phosphorylation to anaerobic glycolysis, ATP production is reduced, intracellular lactate accumulates, intracellular pH is reduced during ischemia, mitochondrial membrane permeability transition pore (mPTP) is inhibited, and intracellular and mitochondrial $Ca^{2+}$ overload are caused. During reperfusion, oxygen and nutrient supply is sufficient, and mitochondria are reactivated, resulting in further mitochondrial $Ca^{2+}$ overload and oxidative stress. Combined with the rapid pH recovery, this induces mPTP to open, causing the mitochondrial membrane potential to change. Subsequently, a large amount of cytochrome C is released into the cytoplasm from the mitochondria, and mitochondrial biogenesis decreases, inducing excessive contraction of cardiomyocytes, triggering mitochondrial dysfunction and cardiomyocyte death (Fig. 1) (*Ramachandra et al., 2020*; *Hernandez-Resendiz et al., 2020*).
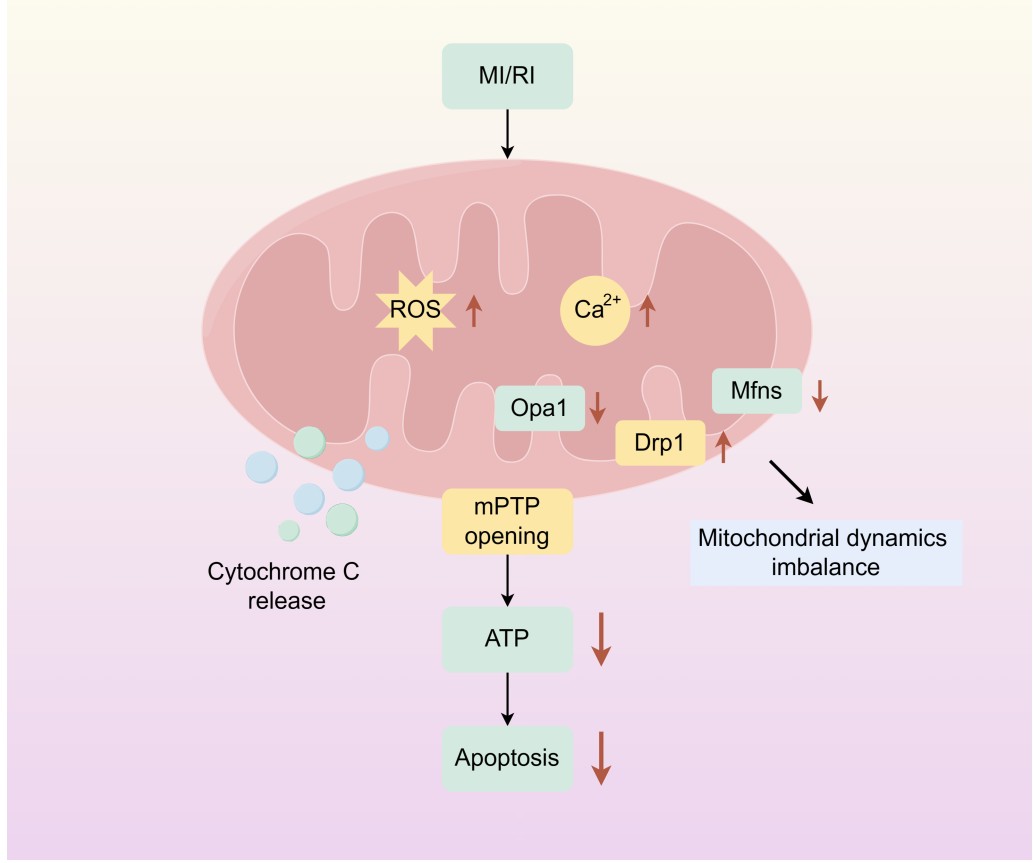

**Figure 1  Mitochondrial damage caused by myocardial ischemia-reperfusion.** The simplified scheme of apoptosis caused by mitochondrial damage during myocardial ischemia-reperfusion injury. MIRI, myocardial ischemia-reperfusion; ROS, Reactive oxygen species; mPTP, Mitochondrial membrane permeability transition pore. Figure created using Figdraw.

### Mitochondrial dynamics and MI/RI

Over the last twenty years, there has been a growing recognition of mitochondrial dysfunction as one of primary pathogenic mechanism in a variety of diseases, such as cardiovascular diseases, and numerous studies have shown that disturbances in mitochondrial dynamics are particularly evident in MI/RI (*Forte et al., 2021*). Apoptosis is linked to the maintenance of mitochondrial morphology and function through the crucial process of mitochondrial dynamics (*Bertholet et al., 2016*). During MI/RI, there are abnormal changes in mitochondrial dynamics, primarily characterized by reduced fusion and increased fission, resulting in membrane potential collapse and massive production of ROS, which leads to cell damage or death (*Xu et al., 2017*).

The primary mediators of mitochondrial fusion are optic atrophy 1 (Opa1) which is positioned on the inner mitochondrial membrane, and mitofusin 1(Mfn1), mitofusin 2 (Mfn2) which are placed on the outer mitochondrial membrane and mediates

mitochondrial inner membrane fusion (*Ong et al., 2015*). Our group has found that during MI/RI Opa1 expression is down-regulated, oxidative stress is increased, and apoptosis is increased, while $\kappa$-OR activation can significantly up-regulate the expression of Opa1 during MI/RI, inhibit the occurrence of oxidative stress, reduce apoptosis, and protect the myocardium (*Wang et al., 2020*). *Zhang et al. (2019)* also demonstrated that cardiomyocyte-specific knockdown of Opa1 caused cardiomyocyte dysfunction, which was thought to be associated with phosphorylation of adenosine monophosphate-activated protein kinase (AMPK). *Du et al. (2022)* discovered that the induction of I/R or H/R induced a reducion in the expression of mRNA levels of mitochondrial fusion factors such as Mfn1, Mfn2, along with an increase in ROS, resulting in irreversible myocardial injury.

Mitochondrial fission is mainly regulated by translocation of dynamic protein-related protein 1 (Drp1) from the cytoplasm to the outer surface of mitochondria, separating intracellular dysfunctional mitochondria from mitochondria with respiratory activity with the help of mitochondrial fission 1 protein (Fis1), mitochondrial fission factor (Mff), mitochondrial dynamics protein 49 kDa (Mid49), and mitochondrial dynamics protein 51 kDa (Mid51) (*Wang et al., 2020*; *Picca et al., 2018*). Our group has found that MI/RI can induce abnormally increased mitochondrial fission, and $\kappa$-OR activation can significantly inhibit excessive mitochondrial fission and reduce apoptosis, and this effect is achieved by regulating Drp1 phosphorylation and translocation activation. *Ma et al. (2017)* documented that Drp1 expression was up-regulated and involved in mitochondrial damage under hypoxia. *Jin et al. (2018)* found that Mff protein was significantly up-regulated after H/R, which triggered mitochondrial fission. And this excessive fission then initiated the caspase 9-associated pathway for mitochondrial apoptosis, ultimately causing massive cell death.

## Mechanism of $\kappa$-OR activation in reducing mitochondrial dysfunction and protecting myocardium

In this paragraph we focus on the $\kappa$-OR/AMPK/GSK-3$\beta$ pathway. Present studies generally support that activation of AMPK, the cellular energy regulator, and its downstream signalling pathway involving glycogen synthase kinase-3$\beta$ (GSK-3$\beta$) is a natural compensatory mechanism during MI/RI, which is an endogenous compensatory mechanism that can inhibit cardiomyocyte apoptosis, reduce myocardial oxidative stress, and alleviate myocardial injury (*Lu et al., 2019*; *Ma et al., 2010*; *Wang et al., 2022a*; *Wang et al., 2022b*). Our group has found that U50,488H can activate the AMPK/GSK-3$\beta$ signalling pathway through $\kappa$-OR, then relieve mitochondrial dysfunction during MI/RI, and reduce myocardial injury (*Tian et al., 2019*). In the treatment group injected intraperitoneally with U50,488H (2 mg/kg) before reperfusion, compared with the I/R model group, the ATP yield and mitochondrial complex level were significantly increased, the opening of myocardial Mito-mPTP and the releaseof cytochrome c were significantly reduced, the expression of myocardial mitochondrial biosynthetic proteins PGC-1[3] and NRF-1 was up-regulated, mitochondrial function was improved, myocardial apoptosis was reduced, and MI/RI was alleviated. And the above results were reversed when $\kappa$-OR, AMPK and GSK-3$\beta$ were blocked, respectively, which suggest that the effect of U50,488H in protecting ischemic

myocardium is related to the AMPK/GSK-3$\beta$ signalling pathway. Furthermore, the research group found that $\kappa$-OR activation can significantly up-regulate the phosphorylated protein expression of AMPK$\alpha$ and GSK-3$\beta$ during ischemia-reperfusion compared with the I/R group. In order to clarify the relationship between $\kappa$-OR and AMPK and GSK-3$\beta$, cells were transfected with AMPK$\alpha$ siRNA in the H/R model, and it was found that after down-regulation of AMPK, mitochondrial fission increased, Mito-mPTP increased, and phosphorylation of AMPK[3] and GSK-3$\beta$ decreased, and the protective effect of U50,488H on ischemic myocardium was inhibited. This demonstrates that activation of $\kappa$-OR can activate the AMPK/GSK-3$\beta$ signalling pathway, improve mitochondrial morphology and function, and protect ischemic myocardium (*Tian et al., 2019*).

We will focus on the $\kappa$-OR/PI3/AKT signalling pathway in this section. PI3K could phosphorylate the D3 hydroxyl group of the phosphatidylinositol lipid ring as a lipid kinase. AKT is a downstream signalling enzyme that is ubiquitously expressed at high levels in the brain, heart and lungs, and both activities are regulated by G protein-coupled receptors (*Ghafouri-Fard et al., 2022*). PI3K/AKT signalling activates downstream effectors and plays a crucial role in regulating cell cycle transition, growth, and proliferation (*Shi et al., 2019*). Our group has found that U50,488H can activate the PI3K/AKT pathway through $\kappa$-OR and reduce mitochondrial damage when MI/RI occur (*Wang, 2019*). The results showed that activation of $\kappa$-OR by U50,488H significantly promote the expression of phosphorylation of PI3K (p-PI3K) and phosphorylation of AKT (p-Akt), restored mitochondrial function, reduced excessive mitochondrial fission, and inhibited apoptosis in ischemia-reperfusion myocardium compared with the I/R group, this effect can be blocked by nor- BNI, a $\kappa$-OR blocker, and PI3K blocker. Transmission electron microscopy showed that activation of $\kappa$-OR inhibited the decrease of mitochondrial cross-sectional area and the increase of mitochondrial fission induced by I/R compared with the I/R group. Myocardial mitochondria were observed by confocal laser scanning microscopy, which was consistent with the results of electron microscopy. The above effects can be blocked by $\kappa$-OR, PI3K, and AKT blockers, respectively, suggesting that the effect of activating $\kappa$-OR to improve mitochondrial function was related to PI3K/AKT signalling pathway. Further examination of mitochondrial fusion split protein expression showed that activation of $\kappa$-OR up-regulated Drp1 (Ser616) expression and downregulated phosphorylated Drp1 (Ser637) expression compared with the I/R group, while other fusion split proteins did not change significantly, and inhibitors of $\kappa$-OR, PI3K, and Akt blocked the effect of U50,488H activation of $\kappa$-OR in regulating Drp1 phosphorylation. $\kappa$-OR exerts a protective effect on ischemic myocardium by regulating the phosphorylation of Drp1 through the $\kappa$-OR/PI3/Akt signalling pathway, inhibiting mitochondrial fission, and reducing apoptosis. In addition, this pathway can also be involved in the pathogenesis of a variety of heart diseases by regulating mTOR, FoxO1/3, and nitric oxide synthase (NOS), however, it is not fully understood whether its downstream signalling molecules are also regulated by $\kappa$-OR (*Deng & Zhou, 2023*).

In this section, our focus is on the $\kappa$-OR/STAT3/Opa1pathway. The research conducted by our group has found that the activation of $\kappa$-OR can enhance the production of the mitochondrial fusion protein Opa1 in the myocardium of mice. This activation inhibits cell apoptosis and oxidative stress during MI/RI, alleviating MI/RI. It is believed that this is associated with the activation of STAT3 (*Wang et al., 2020*). STAT3 is a nuclear transcription factor involved in the regulation of cell cycle, cell survival, and immune responses and plays a role in maintaining mitochondrial function (*Comita et al., 2021*). Observing mitochondrial morphology in mouse heart tissue and primary neonatal rat cardiomyocytes revealed that activation of $\kappa$-OR significantly inhibited excessive mitochondrial fission, decreased the average mitochondrial size, and reduced the number of mitochondria per ¿m compared with the I/R group. Nor-BNI, a blocker of $\kappa$-OR, and STAT3, a blocker of STAT3, significantly inhibited this effect, suggesting that $\kappa$-OR regulates mitochondrial dynamics associated with STAT3. Furthermore, it was found that p-STAT3 (Tyr705) and Opa1 protein expression was up-regulated in both myocardial tissues and cells after $\kappa$-OR activation compared with the I/R group, and nor-BNI. In addition, activation of STAT3 by activating $\kappa$-OR was able to reduce the occurrence of MI/RI-induced oxidative stress, it was found that compared with the sham operation (Sham) group, lactate dehydrogenase (LDH) and creatine kinase-MB isoenzyme (CK-MB) levels in the serum of mice were significantly increased, reactive oxygen species (ROS) level and lipid peroxide malondialdehyde (MDA) content in the myocardial tissue of mice were increased, and superoxide dismutase (MnSOD) activity was attenuated, after $\kappa$-OR activation, LDH and CK-MB levels were decreased, ROS and MDA contents were decreased, MnSOD activity was increased, and myocardial oxidative stress was inhibited compared with the I/R group, while blockers of $\kappa$-OR and STAT3 reversed this result, demonstrating that $\kappa$-OR through activating STAT3, increased the expression of OPA1, promoted mitochondrial fusion, reduced oxidative stress, and anti-MI/RI (*Wang et al., 2020*).

In this section we are going to focus on the $\kappa$-OR regulates translocation activation of Drp1. With the I/R model, our group found that administering U50,488H 10 min before ischemia could inhibit excessive mitochondrial fission, promote mitochondrial fusion, and improve mitochondrial dynamics disorders (*Gao et al., 2023*). This is because of the fact that activation of $\kappa$-OR by U50,488H inhibits Drp1 mitochondrial translocation, alleviates abnormal mitochondrial fission, and then reduces apoptosis and exerts anti-MI/RI effects. By isolating cytoplasmic and mitochondrial proteins in myocardial tissue, the detection of Drp1 expression revealed that there was no significant alteration in cytoplasmic Drp1 expression in each group, however, compared with the Sham group, Drp1 content in myocardial mitochondria was significantly increased in the I/R group, and decreased after activation of $\kappa$-OR compared with the I/R group, suggesting that activation of $\kappa$-OR inhibits mitochondrial translocation of Drp1 in myocardium. In order to further verify the effect of activation- $\kappa$-OR on the mitochondrial translocation of Drp1, the expression of Drp1 in mitochondria and cells was observed by immunofluorescence. Compared with Sham group, H/R can increase the mitochondrial translocation of Drp1 in cardiomyocytes, and activation-OR can inhibit the mitochondrial translocation activation of Drp1 induced by
H/R. Further study showed that activated $\kappa$-OR significantly inhibited MiD51 expression compared with H/R group. Furthermore, small interfering SiRNA was used to knock out neonatal rat cardiomyocytes MiD51. Immunofluorescence results showed that, compared with the H/R group, the knockdown of MiD51 alleviated mitochondrial division of cardiomyocytes and inhibited the activation of Drp1 mitochondrial translocation. These results indicate that $\kappa$-OR can down-regulate the expression of MiD51, inhibit the activation of translocation and Drp1, reduce mitochondrial division and apoptosis, and play an anti-MI/RI role.

## AMPK/Akt/eNOS signalling activation

AMPK is an important energy sensor and metabolic regulator (*Hardie, 2008*). When MI/RI occurs, ATP decline will induce AMPK activation and promote AMPK phosphorylation (*Kemp et al., 1999*). AKT can also contron (*Mullonkal & Toledo-Pereyra, 2007*). AKT can promote glucose uptake through phosphorylation and increase glycolysis through phosphorylated hexokinase (*Chang, Chiang & Saltiel, 2004*; *Miyamoto, Murphy & Brown, 2008*). *Lee et al. (2015)* reported that AMPK and AKT can reduce lactic acid accumulation and cell death when synergically activated by pyrrole alkyl cafeamide. The nitric oxide synthase (NOS) family comprises neuronal, inducible, and endothelial NOS (n/i/e NOS), which are responsible for *in vivo* nitric oxide (NO) synthesis and play distinct roles in cardiovascular disease development (*Ravikumar et al., 2021*). Among these isoforms, NO derived from eNOS exerts cardioprotective effects by reducing blood pressure, inhibiting atherosclerosis formation, and preventing myocardial infarction (*Razavi, Hamilton & Feng, 2005*). Additionally, a separate study demonstrated that eNOS knockdown significantly exacerbated myocardial ischemia/reperfusion injury; impaired eNOS function due to pressure overload resulted in greater left ventricular hypertrophy and dysfunction compared to wild-type mice (*Lee et al., 2016*). In another study conducted by our group, we found that $\kappa$-OR activation plays a cardioprotective role through the AMPK/AKT/eNOS signaling pathway (*Zhang et al., 2018*). In this study, cardiac protection was assessed in rats, myocardial infarction and serum cTNT levels were measured. the results showed that Sham group that silk was drilled underneath the LAD (left anterior descending) but the LAD was not ligated compared with I/R group, cardiac function decreased, dead area and cTNT level increased significantly. Compared with I/R group, the cardiac function of U50,488H+I/R group was significantly improved, myocardial infarction size and cTNT level were decreased. However, nor-BNI, compound C (an AMPK inhibitor), Akt inhibitor, and L-NAME (an eNOS inhibitor) eliminated the effects of U50488. When AICAR was used as the activator of AMPK, activation of AMPK was found to have the same effect as activation of $\kappa$-OR, but this beneficial effect was eliminated by compound C, Akt inhibitor, and L-NAME, suggesting that activations of $\kappa$-OR and AMPK may potentially regulate the Akt and eNOS signaling pathways. Further studies showed that compared with the sham operation group, I/R inhibited the phosphorylation expression of AMPK, AKT and eNOS, while U50,488H treatment could up-regulate the phosphorylation expression of the three and this effect was blocked by nor-BNI, indicating that the downstream signaling pathway molecules of kor included AMPK, AKT and eNOS. Compared with I/R, AICAR+I/R

up-regulates phosphorylation of AKT and eNOS, which is inhibited by compound C, suggesting that AKT and eNOS may be downstream targets of AMPK. *In vitro* experiments, it was further found that L-NAME could not eliminate the regulation of U50,488H and AICAR on AKT phosphorylation at H/R, thus indicating the upstream and downstream relationship. In summary, AKT is the downstream signaling molecule of AMPK, while eNOS is the downstream signaling molecule of AKT. Both are regulated by $\kappa$-OR activation. $\kappa$-OR activation can improve cardiac function, decrease myocardial infarction size and serum cTnT level in rats with myocardial I/R injury. These cardiac protective effects may be achieved through the AMPK/Akt/eNOS signaling pathway (*Wong, Lee & Tai, 1990*; *Chien et al., 1994*).

In addition, it has been reported that peripheral opioid kappa1 receptor activation can reduce infarction and prevent reperfusion heart injury; *Peart et al. (2008)* found that compared with ischemia-reperfusion rats, activating $\kappa$-OR 10 min before ischemia or 5min before reperfusion can significantly reduce myocardial infarction size, and this protection involves the activation of PI3K pathway and mitochondrial $K^+$ channel. *Lishmanov et al. (2003)* documented that activating $\kappa$-OR could reduce MI/RI induced arrhythmia and alleviate MI/RI by activating $K^+$ channel (*Popov et al., 2021*; *Peart et al., 2008*). The above effects are consistent with the results of our group's research.

## DISCUSSION AND FURTHER PERSPECTIVES

### Main interpretation

The current state-of-the-art revascularization techniques in the clinical management of acute myocardial infarction include percutaneous coronary intervention and surgical coronary artery bypass grafting. Cardio-cerebrovascular diseases remain the main killer worldwide (*Li et al., 2021*; *Xin et al., 2019*; *Li et al., 2017*; *Zhang et al., 2017*; *Zhang et al., 2021*; *Liu et al., 2016*). Despite their effectiveness in preventing loss of systolic function, reducing infarct size, and improving clinical outcomes through early recanalization, these modalities do not mitigate the reperfusion injury caused by recanalization (*Algoet et al., 2023*). *Rossello, Pocock & Julian (2015)* concluded that drugs such as $\beta$-blockers, angiotensin-converting enzyme inhibitors, and statins significantly diminish the extent of MI/RI-induced myocardial infarction. However, none of these drugs have been translated into clinical use yet; they are still at the animal experimentation stage (*Rossello, Pocock & Julian, 2015*). Some studies have demonstrated that Danlou Tablet, a traditional Chinese medicine formulation, reduces the expression of apoptotic proteins Bim and PUMA during MIRI by phosphorylating AKT and FoxO3a to enhance cardiac function (*Li et al., 2023*). Furthermore, Trans sodium crocetinate (TSC), a derivative compound derived from carotenoid crocetin attenuates ischemia/reperfusion-induced apoptosis in oxidatively stressed cardiomyocytes through activation of the SIRT3/FOXO3a/SOD2 signalling pathway. It exhibits an anti-MIRI effect with antimuscarinic properties (*Chang et al., 2019*).

Myocardial protective measures, such as myocardial ischemic preconditioning, postconditioning, remote ischemic preconditioning, and exercise preconditioning, exhibit

varying degrees of association with cardiac opioid receptors (*Melo et al., 2018*). In this study, we summarized the protective effect and mechanism of U50,488H on ischemic myocardium by activating $\kappa$-OR during MI/RI, and clarified that activating $\kappa$-OR can inhibit mPTP opening, reduce cytochrome C release, and then reduce apoptosis by regulating mitochondrial dynamics, reducing oxidative stress level. The mechanism of the protective effect of U50,488H activating $\kappa$-OR in MI/RI was also described, involving the following pathways, $\kappa$-OR activation stimulates AMPK/GSK-3$\beta$ signalling pathway, reducing Mito-mPTP opening, inhibiting apoptosis, and improving mitochondrial morphology and function, activates PI3K/AKT signalling pathway, regulating Drp1 phosphorylation and dephosphorylation, and inhibiting excessive mitochondrial fission, promotes STAT3 phosphorylation, up-regulating Opa1 expression, and promoting mitochondrial fusion, inhibits Drp1 translocation activation, down-regulating MiD51, and maintaining the stability of mitochondrial dynamics (Fig. 2).

## LIMITATIONS

Through the continuous study of MI/RI mechanism and therapeutic measures, various pathogenic mechanisms have been elucidated, but its injury mechanism is complex and extensive, and has not yet been fully understood. Current studies confirm that MIRI leads to mitochondrial dysfunction, $Ca^{2+}$ overload, oxidative stress, autophagy, apoptosis, necrosis, iron death and inflammation; however, a comprehensive explanation of the underlying mechanism remains incomplete. Iron death is a mode of cell-regulated death caused by iron-dependent lipid peroxidation which is mainly manifested in the shrinkage of mitochondria with the reduction or disappearance of mitochondrial cristae and the increase of membrane density (*Deng, 2021*). It is important to note that our study demonstrates that agonistic $\kappa$-OR alleviates ischemia-reperfusioninduced myocardial injury by mitigating mitochondrial dysfunction; however, further investigation is required to determine whether agonistic $\kappa$-OR is associated with iron death. Moreover, our study focuses on exploring whether $\kappa$-OR regulates additional proteins involved in mitochondrial dynamics to protect the myocardium. Specifically we primarily examine five proteins associated with mitochondrial fusion and fission. Additionally, the aforementioned findings have not been validated in animal models featuring myocardial knockout or overexpression of the $\kappa$-OR vector due to incomplete construction of transgenic mice during previous stages.

## CONCLUSION

In this paper, the activation of $\kappa$-OR can promote mitochondrial biosynthesis, maintain mitochondrial morphology and function, inhibit oxidative stress and reduce apoptosis, thereby reducing MI/RI. The effects of $\kappa$-OR activation and $\kappa$-OR/AMPK/GSK-3$\beta$ were summarized $\kappa$-OR/PI3K/AKT, $\kappa$-OR/STAT3/Opa1, and $\kappa$-OR regulate Drp1 translocator activation and $\kappa$-OR/AMPK/AKT/eNOS correlation, these findings indicate promising therapeutic implications.

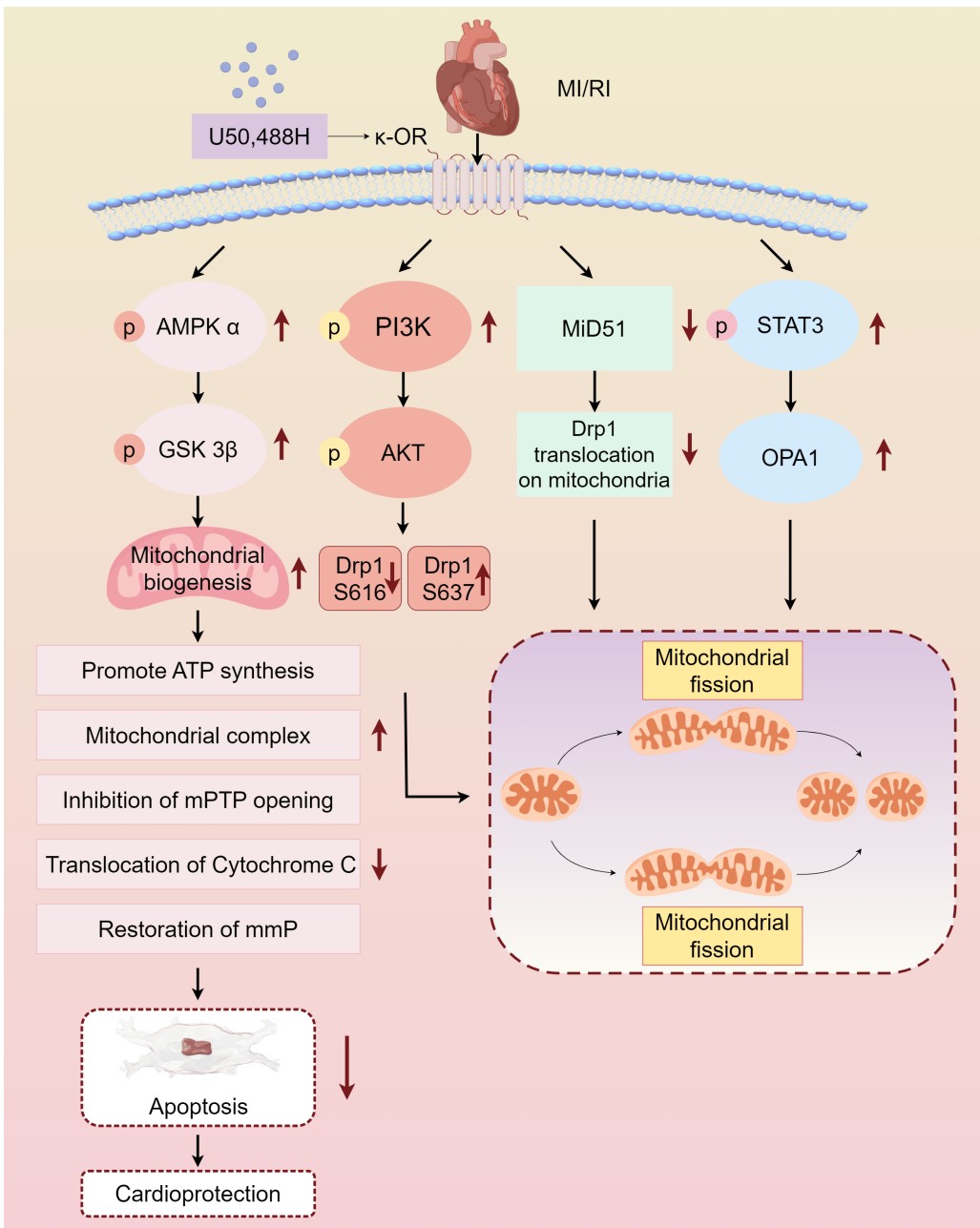

**Figure 2** **Main pathways of κ-OR regulation in MI/RI.** A simplified scheme for κ-OR signal transduction regulation in MI/RI. AMPK/GSK-3 β pathway is in light pink, PI3K/AKT pathway is in rose red, MiD51 pathway is in green, STAT3/OPA1 pathway is in blue and mitochondrial dynamics diagrams are in yellow. mmP: mitochondrial membrane potential, MI/R, Myocardial ischemia-reperfusion; mPTP, Mitochondrial membrane permeability transition pore. Figure created using Figdraw.

## ACKNOWLEDGEMENTS

We thank Home for Researchers editorial team for language editing service.

### Funding

This work was supported by the National Natural Science Foundation of China (No. 81770243) and the Key Research and Development Project of Shaanxi Province (No. 2023-YBSF-373). The funders had no role in study design, data collection and analysis, decision to publish, or preparation of the manuscript.

### Grant Disclosures

The following grant information was disclosed by the authors:
National Natural Science Foundation of China: No. 81770243.
Key Research and Development Project of Shaanxi Province: No. 2023-YBSF-373.

### Competing Interests

The authors declare there are no competing interests.

### Author Contributions

- Wen Zhang conceived and designed the experiments, performed the experiments, analyzed the data, prepared figures and/or tables, and approved the final draft.
- Qi Zhang conceived and designed the experiments, analyzed the data, prepared figures and/or tables, and approved the final draft.
- Yali Liu performed the experiments, authored or reviewed drafts of the article, and approved the final draft.
- Jianming Pei performed the experiments, authored or reviewed drafts of the article, and approved the final draft.
- Na Feng performed the experiments, authored or reviewed drafts of the article, and approved the final draft.

### Data Availability

   This article is a literature review.

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
