# Peer review of "Novel roles of κ-opioid receptor in myocardial ischemia-reperfusion injury"

_PeerJ, doi:10.7717/peerj.17333_

## Round 0.1 · original submission · Major Revisions

As you can see, both reviewers raised several serious concerns. Please address all queries of both reviewers and amend the manuscript accordingly.

**Language Note:** The review process has identified that the English language must be improved. PeerJ can provide language editing services - please contact us at [email protected] for pricing (be sure to provide your manuscript number and title). Alternatively, you should make your own arrangements to improve the language quality and provide details in your response letter. – PeerJ Staff

·

Basic reporting

The topic of the review was not novel, and many previous published reviews addressed this topic.

The abstract was not clear, and the aim of this review in the abstract was different. In the abstract, the aim (line 29) was to evaluate the mechanisms of the k-opioid receptor activated by its exogenous selective agonist U50,488H, whilst in the manuscript ( line 58), the aim was to discuss and elaborate on the regulation of k-OR on mitochondria when MI/RI occurs !!

The article lacks a clear academic writing style. The authors were not choosing their words precisely, not constructing their sentences carefully and not using grammar correctly. Furthermore, some statements were unrelated to the topic, such as in line 273.

Many statements in the review lacked the references, such as in lines 69,77, 81, and many others. Moreover, the authors often mentioned their experimental studies without any references, such as lines 147, 162,233 and others.

The article used some terms unrelated to the academic field, such as acute heart attack instead of acute myocardial infarction and recanalization instead of revascularization.

In the reference line (410-411) ( Cheng, L., Ma, S., Wei, L.X., Guo, H.T., Huang, L.Y., Bi, H., Fan, R., Li, J., Liu, Y.L., and Wang, Y.M. 2007. antiarrhythmic, ventricular arrhythmia. Heart & Vessels 22:335-344.) I could not find this reference in PubMed or even in Google.

Experimental design

The review does not discuss studies that do not agree that the k-opioid receptor is cardiac protective in reperfusion injury. Previous studies demonstrated that the role of κ-opioid receptors in cardioprotection is more controversial.

The author stated that myocardial reperfusion injury is a cardiovascular disease (line 51). That was not true. Myocardial reperfusion injury is a pathophysiological mechanism that occurs in many organs (such as the Brain, heart, and liver), and it is not a disease.
In the main interpretation section, line 321, the authors used the wrong term (systolic myocardial mass instead of systolic function).

Validity of the findings

The authors mentioned that Ibanez et al., 2018 (drugs such as B-blockers, angiotensin-converting enzyme inhibitors, and statins significantly diminish the extent of MI/RI-induced myocardial infarction. However, none of these drugs have been translated into clinical use yet; they are still at the animal experimentation stage).
However, Ibanez et al.'s 2018 article was the current European Society of Cardiology guidelines for managing acute myocardial infarction and never mentioned that ( I did not find that in the article).

Furthermore, many experimental and clinical trials evaluated myocardial reperfusion Injuries not mentioned and used many treatments such as xenon and targeted hypothermia.

In the limitation section, line 355, the authors stated that (further investigation is required to determine whether agonistic k-OR is associated with iron death) This statement was unclear, and what was meant by iron death!!
The conclusion had a big paragraph with simple conclusions

Reviewer 2 ·

Basic reporting

1) The English in the text should be improved.

2) The text contains typos:
line 132 andmediates;
line 157 theĸ-OR;
line 297 that kor downstream signaling, KOR is an abbreviation that did not mentioned before;
line 290 sham group. It is not clear what is Sham group, that did not explained in text)

3) Authors should avoid phrase repeats:
line 24 accumulation, accumulation
line 38 among Chinese residents and line 39 among Chinese residents
line 85 activating k-OR and line 87 activating k-OR

Experimental design

4) The section Survey Methodology needs to be removed because it is self-evident.

The phrase on line 82 Our group has been studying the protective effect of …..for a long time, needs to be removed because the quality of published data matters, not the length of the study.

Validity of the findings

5) No references to Ma et al., 2010 and 2017 in the text of the paper.

6) The results and conclusions in the text of the paper have to be supported by references:
from line 186 to line 204 need to indicate a reference to mentioned research
from line 232 to line264 need to indicate a reference to the mentioned research
line 270, there is no reference to NO cardioprotective effects
line 276, there is no reference to AMPK and AKT cardioprotective effects
from line 276 to line 315, you need to indicate references to the mentioned research/research. Two pages without any references.

Additional comments

From my point of view, your most important omission is that you wrote whole review paper about mechanism of activating k-opioid receptors and effects on mitochondtial dynamics and reduce MI/RI in great extent as discussion resultls from only two your experimental papers (Tian et al., 2019 and Wang et al., 2020)

I recommend shortening the text from line 232 to line 264 and from line 276 to line 315 and including a discussion about published data in the same research field from other groups.

---

## Round 0.2 · Minor Revisions

Please address the remaining issues pointed out by the reviewer and revise the manuscript accordingly.

Reviewer 2 ·

Basic reporting

The authors corrected the text and took into account the majority of suggestions from reviewers. However, the sloppy writing causes concerns. Please see the indicated points below.

Line 57
Why The Brain with a capital “The”? Do you mean cerebral ischemia-reperfusion? It is incorrect to write that myocardial reperfusion injury occurs in the brain: “Myocardial reperfusion injury is a pathophysiological mechanism that occurs in many organs (such as the Brain, heart, and liver)”.

Experimental design

There is no reference for data From Line 276 to 294 in the text. It is not clear what article is being referred to.

Validity of the findings

From line 454 to 461 text contains typos.

---

## Round 0.3 · accepted · Accept

Thank you for addressing the remaining concerns of the reviewer and providing the revised version of your manuscript. I am glad to inform you that your manuscript is acceptable now.